# The Drivers and Barriers of Corporate Social Responsibility: A Comparison of the MENA Region and Western Countries

**Amin Alizadeh**

Department of Educational Administration & Human Resource Development, College of Education, Texas A&M University, College Station, TX 77843, USA; ameen59@tamu.edu.com

**Abstract:** (1) Although numerous articles have been published to address the drivers or barriers of corporate social responsibility (CSR), some parts of the world have received less attention. In this study, I reviewed the literature from 2010 to 2021 to identify drivers and barriers of CSR in the Middle East and North Africa (MENA) region and compare them with the findings in Western countries. (2) Methods: For this study, I used a structured literature review method. By setting the inclusion and exclusion criteria, only 28 articles remained from the selected database. (3) Results: The findings revealed that some CSR drivers, such as leadership styles, profitability, reputation, moral commitment, and environmental conservation, are common in both regions. There are also some differences between CSR drivers, for example, religious beliefs, low concentration of ownership, and company characteristics are some of the drivers in the MENA region. Maintaining social license to operate, and avoiding the risks of community opposition, pressure from the government, and consumer demand tend to be more important in Western countries. Common barriers in both regions are lack of financial resources, cost, lack of CSR knowledge and awareness, and ownership concentration. This review also highlighted that lack of law enforcement, lack of stakeholder communication, lack of management commitment, lack of interests, corruption, and financial debts are some of the barriers of CSR addressed in the MENA region, whereas cost/benefit ratio, lack of customer interest, and lack of scientific frameworks are special barriers in Western countries. (4) Conclusions: Although researchers in Western countries have more focus on the energy sector, there is a lack of research about the drivers and barriers of CSR in the MENA region in several industries, including oil and gas.

**Keywords:** CSR drivers and barriers; corporate social responsibility; MENA; Middle East; North Africa; North America; Australia; Europe; Western countries

## 1. Introduction

Across the globe, CSR is a famous and widespread concept. From political leaders and corporations to marketing and human resource practitioners, people have discussed this topic in detail. In this era, anyone who watches television or reads news will see multiple reports about for-profit companies' social activities, which mainly relate to protection and welfare of the environment and civil society. CSR is the act of adding ethical and moral responsibilities in an organization's goals and decision-making strategies [1]. In 1999, Carroll declared that large corporations have great decision-making power to influence the lives of everyone in society, and those decisions should be made by company leaders, based on societal values [2]. Nowadays, CSR activities have become a wide range of programs that highly impact an organization's core values. These activities are mainly focused on internal/external issues such as employees' work–life balance, employee needs, workplace safety, sustainability, human resource management, the environment, poverty, and community development [1].

Although numerous articles have published to address drivers or barriers of CSR in Western countries, some parts of the world, such as developing countries, have received less attention [3,4]. I also found that there is a lack of updated reviews on CSR drivers

and barriers considering geographical locations. Drivers point to elements that predict the CSR practices and policies, and barriers address obstacles of implementing CSR practices and policies. In this study, I reviewed 28 journal articles from developing and developed countries from 2010 to 2021 and found out how drivers and barriers of CSR from different cultures and industries are similar/different. In this study, the main question is what the drivers and barriers of CSR are as a follow up, I try to define whether CSR drivers and barriers are distinct in different geographical regions. Despite extensive research on CSR, there is still limited literature comparing CSR in Western countries (In this article, Western countries only consist of Australia, North America, and European countries) with rest of the world. This review will offer fresh and updated insights into the growing CSR literature.

*Definition of Corporate Social Responsibility*

Practitioners have used many different terms to address CSR practices, such as social responsibility of business, corporate responsibility, corporate citizenship, business responsibility, corporate social performance, corporate sustainability, corporate supply chain social responsibility, and corporate conscience. Reviewing the literature revealed that there is no single, generally accepted definition of CSR [2]. However, in *Social Responsibilities of the Businessman*, the author proposed one of the earliest definitions for the social responsibility of businesses [5]. According to Carroll, who also proposed one of the most influential definitions of CSR [6], in 1960, Keith Davis provided a leading and crucial definition of CSR [7]. The commission of the European communities—also known as the European Commission—has released CSR definitions every year since 2001 and their definitions have been referred to several times by academic scholars [8–10]. In Table 1, I provided 10 well-known definitions of CSR.

**Table 1.** Definitions of CSR.

| Author (Year) | Definition |
|---|---|
| Bowen (1953) | "The obligations of businessmen to pursue those policies, to make those decisions, or to follow those line of actions which are desirable in terms of the objectives and values of our society" (p. 6). |
| Davis (1960) | "Businessmen's decisions and actions taken for reasons at least partially beyond the firm's direct economic or technical interest" (p. 70). |
| Friedman (1970) | "Corporate social responsibility is to conduct the business in accordance with shareholders' desires, which generally will be to make as much money as possible while conforming to the basic rules of society, both those embodied in law and those embodied in ethical custom" (p. 32). |
| Carroll (1979) | "The social responsibility of business encompasses the economic, legal, ethical and discretionary expectations that society has of organizations at a given point of time" (p. 500). |
| Maignan and Ferrell (2000) | "The extent to which businesses meet the economic, legal, ethical, and discretionary responsibilities imposed on them by their stakeholders" (p. 284). |
| McWilliams and Siegel (2001) | "Situations where the firm goes beyond compliance and engages in actions that appear to further some social good, beyond the interests of the firm and that which is required by law" (p. 117). |
| Rupp et al. (2006) | "Activities, decisions, or policies, that organizations engage in to effect positive social change and environmental sustainability" (p. 537). |
| European Commission (2011) | "The responsibility of enterprises for their impact on society" [11]. |
| Aguinis and Glavas (2012) | "Context-specific organizational actions and policies that take into account stakeholders' expectations and the triple bottom line of economic, social, and environmental performance" (p. 933). |
| Rasche et al. (2017) | "The integration of an enterprise's social, environmental, ethical and philanthropic responsibilities towards society into its operations, processes and core business strategy in cooperation with relevant stakeholders" (p. 6). |

Scholars have come to an agreement on the dimensions of CSR, but the majority of CSR pioneers include stakeholders and social and environmental dimensions as the foundation [12–14]. McWilliams and Siegel affirmed that CSR has two dimensions of social and voluntariness, as social responsibility goes beyond economic and legal dimensions [15]. Other researchers proposed that CSR focus needs to be on community, environment, employees, and customers [16,17]. They excluded the government and economic dimensions of CSR and affirmed that legal requirements are not part of CSR activities [16]. In this study, I follow Rasche et al.'s definition as social, environmental, ethical, and philanthropic dimensions as the main basis for CSR [18].

## 2. Methods

For this study, I used the structured literature review method [19]. Due to pursuing these methods, I identified data points in the literature that inform new or emerging concepts by conducting structured steps of analysis. I contend that in order to develop a better understanding of the CSR drivers, a focus on a different geographic region is needed, as several studies have shown that CSR drivers might vary based on different cultures and countries [20,21]. Although there have been several review articles published related to CSR drivers, there is a lack of reviews that highlight the differences between various cultures and countries.

Journal articles were sourced from ABI/INFORM Complete, Business Source Complete, ERIC (EBSCO), and Web of Science. The keyword search was to identify articles containing "corporate social OR social responsibility OR corporate philanthropy OR corporate citizenship OR CSR" in the title of study and "drivers OR barriers OR antecedents" anywhere in the paper. As CSR is an emerging and evolving topic, I decided to have current and scholarly resources in this evolving field by limiting the search to only empirical articles that had been published after 2010 in the English language. The initial search generated 2282 articles in the selected database.

To identify articles from Western countries, I added an extra row with the name of 26 European countries included in Schengen area, as well as the US, the UK, Canada, and Australia. For the US, I added "U.S.A. OR USA OR United States OR U.S. OR US OR America." For the UK, I added United Kingdom, England, and Scotland. Although different organizations define MENA differently, for this study, I included 19 countries from the Middle East and North Africa by following the WorldAtlas organization. In order to identify MENA-related studies, I added an extra row of country names as follows: "Algeria OR Bahrain OR Egypt OR Iran OR Iraq OR Israel OR Jordan OR Kuwait OR Lebanon OR Libya OR Morocco OR Oman OR Palestine OR Qatar OR Saudi Arabia OR Syria OR Tunisia OR United Arab Emirates OR UAE OR Yemen." The country name had to appear in the abstract. With this additional limit, the total number of articles was reduced. After removing duplicates and reading the abstracts to find appropriate papers, only 9 articles for the MENA region remained.

An inclusive approach would demand the inclusion of all 406 articles from Western countries for the review, but this way was deemed inefficient since I only had 9 articles from the MENA region to compare. Alternatively, I decided to adopt a statistical method to form a representative random sample from the 406 articles. To be 90% certain of being accurate to within +0.1 and −0.1 of the true proportion of all articles, a minimum sample size of 41 articles was needed from this region [22]. I increased the sample to 60 articles to lower the probability of Type II error. In the second step, I followed the inclusion criteria rules (Table 2). Initially, I reviewed only abstracts and skimmed the methodology section to determine relevancy of the articles and in the final step, reviewed the chosen one in depth. I reviewed articles that contained drivers of CSR in depth. A total of 19 articles were selected to compare with 9 articles from the MENA region (See Table 3). By identifying irrelevant papers in the final step, only 28 papers remained in this selection process (Figure 1).

**Table 2.** Inclusion criteria.

| Inclusion Criteria | Rationale |
| --- | --- |
| Title must have corporate social, social responsibility, CSR, corporate philanthropy, corporate citizenship as the focus of the study. | To reduce the chance of collecting unrelated articles. |
| Abstract must show clear indication of drivers or barriers of corporate social responsibility. | The focus of the research is to study drivers and barriers of corporate social responsibility. |
| Country/geographical region needs to be clear in the abstract and has to be among MENA or Western countries. | To compare CSR drivers by categorizing them into two groups based on cultural similarities and geographical locations. |
| Article must be written in English. | English is the dominating research language in the field of corporate social responsibility. |
| Article must be qualitative, quantitative, or mixed method. | To analyze only empirical studies. |
| Only empirical articles from 2010. | To focus on the current drivers and barriers. |

**Table 3.** Number of collected articles for the MENA region.

| Database | Number of Collected Articles | Number of Collected Articles for Western Countries | Number of Collected Articles for the MENA Region |
| --- | --- | --- | --- |
| ABI/INFORM | 495 | 131 | 8 |
| Business Source Complete | 1114 | 148 | 8 |
| ERIC (EBSCO) | 10 | 2 | 2 |
| Web of Science | 663 | 125 | 9 |
| Total First Step | 2282 | 406 | 27 |
| Collected for Analysis | | 19 | 9 |

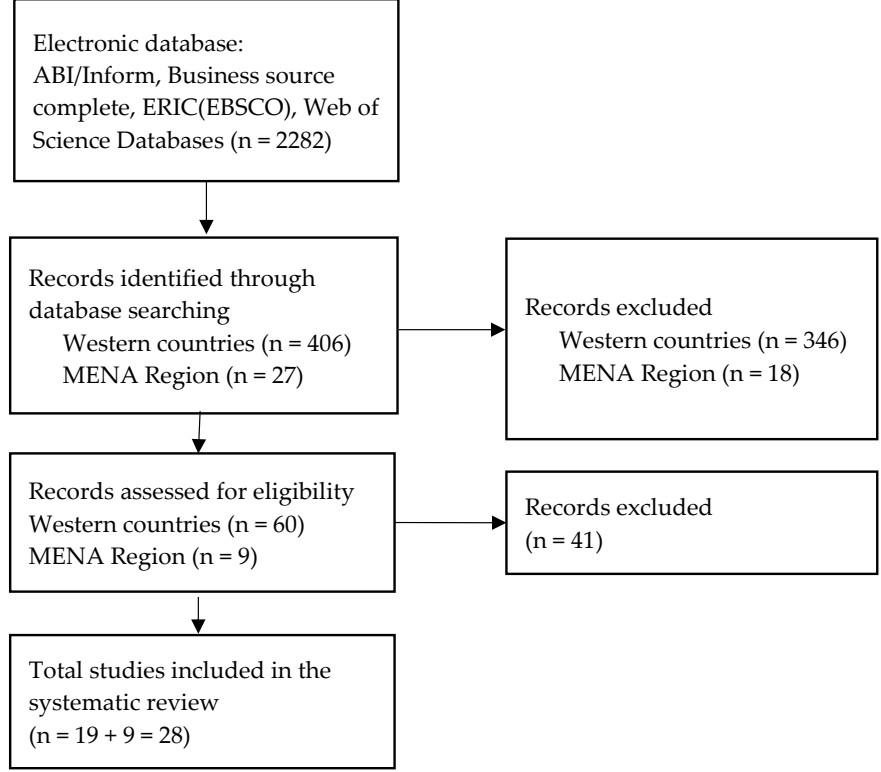

**Figure 1.** Flow chart of the study selection process.

Table 4 presents the distribution of the selected journal articles. Reviewing the selected articles revealed that most studies were published in one of four journals: *Journal of Business Ethics*, *Social Responsibility Journal*, *Sustainability*, and *Corporate Social Responsibility and Environmental Management*. There were 13 qualitative and 15 quantitative studies in this selection and 15 studies collected their data from more than one industry.

**Table 4.** Journal titles and number of articles found.

| Name of Journals | Number of Articles |
|---|---|
| *Journal of business ethics* | 10 |
| *Social Responsibility Journal* | 3 |
| *Corporate Social Responsibility and Environmental Management* | 3 |
| *Sustainability* | 3 |
| *The Extractive Industries and Society* | 1 |
| *The Journal of Asian Finance, Economics, and Business* | 1 |
| *Corporate Reputation Review* | 1 |
| *International Business Review* | 1 |
| *Business Strategy Series* | 1 |
| *International Journal of Entrepreneurial Knowledge* | 1 |
| *Energy Policy* | 1 |
| *Spanish Journal of Finance and Accounting* | 1 |
| *Journal of Financial Reporting and Accounting* | 1 |

The countries or regions included in the CSR studies were identified and are presented in Table 5. The 28 papers covered 18 countries or regions, including both the MENA region and Western (developed) countries. Of these, most papers focused on developed countries; the United States (five) and Australia (three) had the most contribution. Four articles focused on multi-country analysis (e.g., European firms, Nordic countries).

**Table 5.** Distribution of selected papers by country or region.

| No. | Country or Region | Number of Selected Papers |
|---|---|---|
| 1 | United States | 5 |
| 2 | Australia | 3 |
| 3 | Norway | 2 |
| 4 | Sweden | 2 |
| 5 | Iran | 2 |
| 6 | Saudi Arabia | 2 |
| 7 | European firms | 1 |
| 8 | Nordic countries | 1 |
| 9 | The Netherlands | 1 |
| 10 | Spain | 1 |
| 11 | United Kingdom | 1 |
| 12 | North America and Western Europe | 1 |
| 13 | Egypt | 1 |
| 14 | France | 1 |
| 15 | Jordan | 1 |
| 16 | Morocco | 1 |
| 17 | U.A.E. | 1 |
| 18 | U.A.E., Lebanon, and Tunisia | 1 |

*2.1. Corporate Social Responsibility Drivers*

In this section, the goal is to find out what the drivers of CSR are and determine whether geographical location and different culture have any impact on CSR antecedents. By drivers, I aim to discover factors that work as predictors, motives, or forces that lead to CSR implementation in organizations, either willingly or unwillingly. Most companies engage in CSR practices due to institutional and stakeholder pressures [23]. Aguinis and Glavas classified CSR drivers into individual, organizational, and institutional levels and

declared that most articles have studied CSR drivers at the organizational level [23]. Several other scholars divided CSR drivers into three categories: value driven, performance driven, and stakeholder driven [24,25]. The two mentioned studies described that value driven refers to a self-motivated approach and depends on external pressures. They also explained that performance-related drivers depend on the corporation's economic benefits, and also contains the company reputation and image, whereas stakeholder-driven forces depend on stakeholders' demands regarding CSR activities. In previous studies, most scholars categorized CSR drivers into external and internal [26–28], and recently, some articles have attempted to study CSR antecedents by dividing them into three groups of internal, connecting, and external drivers to engage in CSR [29,30].

Drivers of CSR may differ with respect to size of the company and degree of internationalization [31]. Dhanesh confirmed that different cultures may have various drivers for CSR [32]. Mazutis and Zintel reviewed the literature and declared that in general, leader demographic characteristics and personal values are important drivers that could impact CSR formation [33]. One study compared Sweden and Taiwan by using World Value Surveys (WVS) data and revealed that national culture, besides education, plays a critical role in shaping CSR drivers [20]. I reviewed the literature from 2010 to 2021 to explore the main drivers of CSR. I analyzed 28 journal articles that studied CSR antecedents from different countries (see Table 5). Since most of the journal articles focused on studies in North America, Europe, and Australia, my main emphasis was to discover perspectives in the MENA region to make the comparison easier. All of the drivers and barriers of CSR were identified by analyzing organizations' documents or by surveying the companies' employees, business leaders, and managers. Therefore, these findings do not represent consumer perspective. In the following section I address my findings from the Middle East and North Africa (MENA), and North America, Europe, and Australia.

### 2.2. Middle East and North Africa (MENA)

Although the concept of CSR has been analyzed in depth in Western countries, there is a dearth of empirical research in developing nations like the MENA region [34]. One of the latest studies examined the determining factors of CSR in 94 Jordanian companies from the manufacturing and service industries [35]. Their findings revealed that (a) company maturity, profitability, and size have positive impacts on CSR participation; (b) a high percentage of debt to assets is a barrier to Jordanian companies' participation in socially responsible activities; (c) family-owned companies have less incentive to engage in CSR activities; and (d) low concentrated ownership and media coverage are also accounted for as drivers of CSR engagement.

In Morocco, researchers conducted a qualitative study by using semi-structured interviews, focus sessions, and observations in the phosphate mining industry [36]. They affirmed that the most determining driver of CSR was the uprising of mining communities after the Arab Spring. They declared one of the underlying reasons for CSR activities was to defend their reputation and secure the mining production. Another study was conducted in Saudi Arabia and identified the main drivers of CSR as improvement of corporate image and moral commitment [37].

Another study surveyed 105 executives from Iranian manufacturing organizations from various industries [38]. Religious beliefs were found to be one of the main drivers of CSR in Iran. Based on this study, enhanced corporate identity, corporate reputation, and attracting more customers are other important drivers of CSR. Other scholars interviewed 13 experienced Iranian business professionals and affirmed that the main CSR drivers are branding, profitability, community welfare, quality improvement, customer retention, and environmental conservation [39]. Although many interviewees believed that CSR could have positive impacts on companies' reputations, some professionals declared that CSR is a forgotten element of businesses in Iran.

A survey study hired 740 participants from the U.A.E., Lebanon, and Tunisia and introduced participative leadership as a driver of CSR [34]. Later on, Lythreatis et al.

surveyed 752 employees from the trading and service industries and identified servant leadership as a strong antecedent of internal CSR [40].

### 2.3. North America, Europe, and Australia

Laudal (2011) surveyed managers in the Norwegian clothing sector and identified five main drivers and three barriers of CSR in small and medium-sized enterprises (SMEs) and multinational enterprises (MNEs) [31]. He declared that companies tend to follow the practices of leading companies in their field. He called this strategic behavior "following leading companies." Some corporations engage in CSR activity as mutually beneficial partnerships that enhance local and public reputation by showing social and environmental responsibility [31]. Laudal also indicated that CSR could work as a strategic tool for corporations to diminish market risks, create market opportunities, and get involved in public policy decision-making processes. In other words, they may use CSR as a tool to influence or even change public regulations.

A Norwegian study surveyed business students, CEOs, and NGO employees in Norway to compare different perspectives and identify what motivates managers to adopt CSR [41]. Their findings revealed that all three groups were in an agreement that the key drivers for business leaders are branding (to create a positive reputation and brand image), stakeholders (to satisfy different stakeholders), and value maximization (to create long-term value for shareholders). Boukattaya and Omri studied 96 French firms and analyzed the link between board characteristics and CSR. They revealed that women are more sensitive to CSR engagement and ethical challenges, and therefore, board gender diversity has a positive impact on CSR implementation [42].

Chkanikova and Mont identified the drivers and barriers for Swedish food retailers [43]. Pressure from the government and European Union, strengthening reputation and brand name, and consumer demand for greener and healthier products were some of the drivers. They also found many retailers engaged in socially responsible activities such as improving eco-efficiency to reduce operational costs. Another driver was industrial norms, agreements, and certifications that force retailers to follow the agreement for sustainability improvement. Another study in Sweden found that CSR is an outcome of employees' motivation in the workplace [44]. The study also described that CSR implementation in SMEs in the apparel industry is driven by employees' perceptions of moral responsibility for CSR [44].

One study analyzed the gas mining industry in Australia and summarized that the main reason for participation in CSR is to maintain their social license to operate and avoid the risks of community opposition [45]. Another study reviewed the literature to identify how CSR had been implemented in the British construction industry and revealed that good reputation among the public and demand to increase credibility are two main antecedents of CSR in that industry [46]. Bolton et al. studied a British energy company and stated that the organization was engaged in CSR to safeguard its reputation and position itself in ranking indices [47]. Another study examined Royal Dutch Shell, which is one of the biggest corporations in the oil sector and found that profitability is one of the key drivers of CSR programs [48].

Lozano indicated that the drivers of CSR participation could be due to internal or external motivations [30]. He conducted 16 semi-structured interviews with top-level corporate managers and experts in the field. All of the interviewees were working in North American or Western European organizations. His findings revealed that reputation, customer demands and expectations, and regulation and legislation are the main external drivers of CSR, whereas business strategy, corporate culture, cost savings and profitability, environmental performance and climate change adaptation and mitigation, and risk prevention and risk management are the main internal drivers.

Godos-Díez et al. surveyed 101 organizations in Spain and studied the impacts of companies' ownership structure and top management characteristics on CSR implications [49]. They claimed that corporations are framed with the values of their CEOs, when the CEOs

are free to act. Their study revealed that ownership concentration and certain manager characteristics positively influence the implementation of CSR practices. In contrast, two scholars examined 700 European multinational firms from 15 countries and 35 industries and found that more concentrated ownership is a barrier to CSR [50].

Fabrizi et al. studied 597 US firms and found that personal incentives of CEOs have a significant effect on the CSR decisions of companies. For instance, CEOs who are new to the company (or role) and need to gain legitimacy from stakeholders are more likely to engage in CSR activities [51]. Their review also revealed that there are four main CSR drivers in US companies, which are moral obligation, sustainability, license to operate, and company reputation. In a different study, a major Australian bank was analyzed by 11 in-depth interviews and nine meetings and forums. Their findings revealed that leadership styles and values could be institutional drivers of CSR [52].

Jo and Harjoto analyzed 3000 U.S. organizations from the Kinder, Lydenberg, and Domini (KLD) Stats database. They explored the effects of corporate governance on CSR engagement and found that CSR engagement is driven by corporate governance and monitoring systems such as board leadership, board independence, institutional ownership, and analyst following have the most significant and positive effect on a firm's decision to engage in CSR [53]. They emphasized that CSR is a supplement to corporations' activities to adopt successful corporate governance, securing corporations' sustainability through ethical business practices.

Jin and Drozdenko surveyed IT professionals in the United States and explored the relationships between organizational values, organizational ethics, and CSR [54]. Their findings revealed that managers from companies with organic core values (e.g., democratic, open, trusting) have greater levels of social responsibility compared to those with mechanistic values (e.g., structured, regulated, closed). They also found that managers who are more ethical tend to be more socially responsible [54]. Another study in the United States surveyed 466 managers from different industries and declared that organizational core values significantly affect CSR [55]. They also declared that corporations that are more socially responsible earn higher profits in that country.

*2.4. Corporate Social Responsibility Barriers*

There is a need for a certain level of financial freedom before investments in CSR can be expected from SMEs [31]. It is also suggested that implementing CSR requires the capacity to devote time, knowledge, and facilities to an area where no immediate returns on investment can be expected. Laudal called the two mentioned barriers "insufficient cost/benefit ratio" and "external control", which are two factors for SMEs that make CSR practices an unreachable source of competitive edge. He also addressed "internal control" as one major barrier for large MNEs and found that when the number of suppliers and internal departments increases, the self-interest of each may be in conflict with the CSR objectives and create a barrier for CSR implementation [31].

Chkanikova and Mont addressed several barriers that retailers face to engage in socially responsible actions [43]. They found that lack of governmental leadership to support the transition, lack of financial resources, lack of knowledge and expertise, and high costs of sustainable products are some of the barriers. Globalization and the competitive environment in the retail industry cause customers to search for cheap food or products. This barrier creates a challenging situation for supermarkets to implement sustainability improvements.

One study mentioned that there are several barriers for organizations to implement CSR in Iran [38]. Culturally, people tend to keep their charitable acts and good deeds confidential. Businessowners prefer to participate in socially responsible activities confidentially for religious beliefs and not to show off or take advantage of it. Due to the economic condition in Iran, entrepreneurs are more focused on short-term goals than on developing longer-term strategies such as CSR. The researcher also found that many business owners

believe the government should be more accountable for social responsibility. Lack of knowledge is another barrier that was addressed [38].

El-Bassiouny conducted a qualitative study in Egyptian companies and concluded that the main barriers to CSR implementation are ineffective regulatory and governance systems, high levels of corruption, lack of top management commitment, and insufficient levels of CSR expertise [56]. Alotaibi et al. studied the barriers to CSR implementation within the construction industry in Saudi Arabia [57]. They identified 11 CSR barriers by reviewing the literature and interviewing local CSR experts, then they surveyed 137 respondents from two companies' HR and overarching management departments. Their findings revealed that there are seven main barriers to CSR, such as additional costs, lack of awareness and knowledge, lack of guidelines and coherent strategy, lack of stakeholder communication, lack of law enforcement, lack of training, and unclear project requirements.

Fabrizi et al. claimed that CEOs' and shareholders' monetary interests have a negative effect on CSR implementation [50]. Latapi et al. ried to identify the main barriers of CSR activities in Northern European energy companies [58]. Their research was based on empirical data obtained from interviews involving high-level managers from the largest suppliers of energy in the Nordic region. They addressed seven barriers at the individual level, seven at the organizational level, and three at the institutional level of analysis (see Table 6).

**Table 6.** Drivers and barriers of CSR.

| Author/Year | Drivers/Level | Barriers | Location/Industry/Method |
|---|---|---|---|
| Angus-Leppan et al. (2010) | Leadership styles and values | | Australia/Banking/Qualitative |
| Jin and Drozdenko (2010) | Having ethical managers with organic core values (e.g., democratic, open, trusting) | | US/IT industry/Quantitative |
| Ditlev-Simonsen and Midttun (2011) | Branding, stakeholders, and value maximization | | Norway/Mix industries/Quantitative |
| Bolton et al. (2011) | Safeguarding their reputation, positioning themselves in ranking indexes | | UK/Energy company/Qualitative |
| Ekatah et al. (2011) | Profitability | | The Netherlands/Oil and gas/Qualitative |
| Valmohammadi (2011) | Increasing corporate identity, increasing general reputation, religious beliefs, attracting customers | Lack of knowledge or awareness of CSR, corporations believe the government should be responsible for sustainable development and not-for-profit sector | Iran/Mixed industries/Quantitative |
| Jo and Harjoto (2011) | Corporate governance | | US/Mixed industries/Quantitative |
| Laudal (2011) | The need for good corporate reputation, following leading companies, sensitivity to public and local perceptions, to ward off government regulation (autonomy), geographical spread(risk) | Cost/benefit ratio (capacity) External control (risk) Internal control (risk) | Norway/Clothing industry/Quantitative |
| Dam and Scholtens (2013) | | Ownership concentration | European firms/Mixed industries/Quantitative/ |

**Table 6.** *Cont.*

| Author/Year | Drivers/Level | Barriers | Location/Industry/Method |
|---|---|---|---|
| Ghasemi and Nejati (2013) | Branding, profitability, community welfare, quality improvement, customer retention, and environmental conservation | | Iran/Mixed industries/Qualitative |
| Jin et al. (2013) | Organizational core values | | US/Mixed industries/Quantitative |
| Arli and Cadeaux (2014) | Stakeholder salience (power, legitimacy, urgency) | | Australia/Mixed industries/Qualitative |
| Fabrizi et al. (2014) | Personal incentives of CEOs | CEOs' and shareholders' monetary interests | US/Mixed industries/Quantitative |
| Bashtovaya (2014) | The moral responsibility of doing the right thing | | US/Oil industry/Qualitative |
| Godos-Díez et al. (2014) | Top management characteristics, ownership concentration | | Spain/Mixed industries/Quantitative/ |
| Lozano (2015) | External drivers: reputation, customer demands and expectations, and regulation and legislation. Internal drivers: business strategy, corporate culture, cost savings and profitability, environmental performance and climate change adaptation and mitigation, and risk prevention and risk management. | | North America and Western Europe/Mixed industries/Qualitative |
| Chkanikova and Mont (2015) | Pressure from government and the European Union; stakeholder demands; strengthening reputation and brand; consumer demand; industrial norms; lack of unhealthy food, GMOs, and pesticide use; NGO campaigns; media attention; scientific alerts | Lack of governmental leadership to support retailers, lack of financial resources, lack of knowledge and expertise, more costs for sustainable products, lack of customer interest, lack of scientific framework | Sweden/Food retail industry/Qualitative |
| Curran (2017) | To maintain their social license to operate and avoid the risks of community opposition | | Australia/Gas mining sector/Qualitative |
| Lythreatis et al. (2019) | Participative leadership | | U.A.E., Lebanon, and Tunisia/Mixed industries/Quantitative |
| Sendlhofer (2020) | Employee moral responsibility | | Sweden/Apparel industry/Qualitative |
| Alotaibi et al. (2019) | | Additional costs, lack of awareness and knowledge, lack of guidelines and coherent strategy, lack of stakeholder communication, lack of law enforcement, lack of training, and unclear project requirements | Saudi Arabia/Construction industry/Quantitative |

**Table 6.** *Cont.*

| Author/Year | Drivers/Level | Barriers | Location/Industry/Method |
|---|---|---|---|
| Pinto and Allui (2020) | Improvement of corporate image and moral commitment | Lack of management commitment, lack of investor interests, lack of economic resources, lack of employee competencies | Saudi Arabia/Manufacturing and service industry/Quantitative |
| Mehahad and Bounar (2020) | Uprising of mining communities after the Arab Spring | | Morocco/Mining industry/Qualitative |
| El-Bassiouny (2020) | | Ineffective regulatory and governance systems, relatively high levels of corruption, lack of top management commitment, and insufficient levels of CSR expertise | Egypt/Egyptian companies/Qualitative |
| Lythreatis et al. (2021) | Servant leadership | | U.A.E., Lebanon, and Tunisia/Trading and service industries/Quantitative |
| Ananzeh et al. (2021) | Company characteristics (age, size, profitability, media exposure, and low concentration of ownership) | Financial debts and family ownership | Jordan/Manufacturing and service sectors/Quantitative |
| Boukattaya and Omri (2021) | Board gender diversity | | France/Mixed industries/Quantitative |
| Latapí et al. (2021) | | Individual level: company's negative contribution to society; decision-making based on egocentrism; lack of CSR fit, motivation, and commitment; lack of CSR knowledge and awareness; lack of CSR leadership; lack of organizational support; and negative attitude toward CSR. Organizational level: lack of flexibility and adaptability, lack of integration of CSR into the core business, lack of organizational trust, lack of understanding of the context, limited access to resources, misalignment of the corporate culture, and unfit organizational structure. Institutional level: cognitive, normative, and regulatory barriers. | Nordic countries/Qualitative/energy sector |

## 3. Discussion

This review of the literature indicates that empirical research on CSR drivers and barriers is very limited in the MENA region. For this region, only nine papers were found during the literature review process and four out of the nine studies collected their data without having a focus on any specific industry, whereas the rest analyzed the manufacturing, trading, service, mining, and construction industries. In contrast, researchers in Western countries focused more on the energy sector, with five studies that

concentrated on the energy industry. Most studies in Western countries were qualitative (11 out of 13), whereas only three out of nine studies in the MENA region were qualitative. This study was not able to identify CSR drivers or barriers in some industries, such as healthcare and hospitality, due to the lack of research in those sectors.

In order to fill the gap, researchers need to focus on energy-related companies, including oil and gas corporations in the MENA region. This region is a particularly interesting area to study social responsibility drivers and barriers because of its historical and cultural heritage. The region was the location of Persia and Kemet, which were two of the oldest civilizations in the world with histories of valuing societal concerns. Other countries also have very rich cultural backgrounds; at the same time, MENA has the world's largest oil reserves and is responsible for the vast amounts of toxic air and water pollution and is one of the largest sources of greenhouse gas emissions in the world. There is a need for more focus on this sector to identify the main drivers and barriers of CSR and plan for more engagement in effective activities and a reduction in their impact on the climate crisis.

## 4. Limitations

There are several limitations that have to be considered. This review was limited to only empirical articles that were published after 2010 in the English language. The selection of the publications was also limited to four scientific databases and followed a specific selection criterion. The research was conducted with limited keywords, which left out some articles that did not include the chosen keywords. Although each country has its own laws and legislation, culture, and economic and political situations, I compared the drivers and barriers of CSR without considering any of the mentioned characteristics. The mentioned limitations create potential for future research to consider other strategies and methods within the CSR literature, as well as other primary sources of information, such as analyzing how organizations implement CSR in each country.

## 5. Conclusions and Future Research Recommendations

Based on a systematic literature review of 28 journal articles, this study investigated and compared the drivers and barriers influencing CSR implementation in the MENA region with that in Western countries. My analysis shows that some CSR drivers, such as leadership styles, profitability, reputation, moral commitment, and environmental conservation, are similar in both regions. There are also some differences between CSR drivers; for example, religious beliefs, low concentration of ownership, and company characteristics are some of the drivers of CSR in the MENA region. Maintaining social license to operate, avoiding the risks of community opposition, pressure from the government, and consumer demand tend to be more important in Western countries.

I also found there are common barriers in both regions, such as lack of financial resources, cost, lack of CSR knowledge and awareness, and ownership concentration. This review also discovered that lack of law enforcement, lack of stakeholder communication, lack of management commitment, lack of interests, corruption, and financial debts are some of the barriers of CSR addressed in the MENA region, whereas cost/benefit ratio, lack of customer interest, and lack of scientific frameworks are special barriers in Western countries.

This information is significant because it indicates that there is a gap in CSR literature, and the MENA region in particular has not been studied in depth by researchers; there is room for future research. Although the electricity and transportation sectors as well as oil and gas companies are the main contributors to global emissions, there is a vital need to understand what drives these companies towards socially responsible business practices. Future studies could also conduct comparative studies between different industries to identify the differences and similarities of CSR drivers and barriers in various industries.

The findings also reveal that there are many countries in the MENA region that have not been analyzed. Future research could also consider focusing on the neglected areas. Another key area that seems to be missing in the literature is understanding the role

the COVID-19 pandemic has had on CSR implementation in emerging and developed countries. Due to the fast-growing wave of globalization, it is critical for multi-national companies to discover all of the CSR drivers and barriers in different regions to become more successful internationally. It is also noted that CSR barriers and drivers are varied based on geographical locations.

**Funding:** This research received no external funding.

**Institutional Review Board Statement:** Not applicable.

**Informed Consent Statement:** Not applicable.

**Conflicts of Interest:** The author declares no conflict of interest.

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
