# Peer review of "The Drivers and Barriers of Corporate Social Responsibility: A Comparison of the MENA Region and Western Countries"

_sustainability, doi:10.3390/su14020909_

Round 1
Reviewer 1 Report
It is an adequate work that provides information on drivers and barriers of CSR through a study based on the selection of papers from MENA region. As a consequence, specific drivers and barriers are provided in these countries. So, the knowledge about CSR in emerging countries is completed, together with other existing literature, in comparison with the drivers and barriers in developed countries. As a consequence, a broad vision of CSR is described in this paper.
The work is therefore timely, the author uses a well-structured information gathering methodology and provides some tables that serve to have a clear idea of the contribution.
In short, it is a well-written work with a well-defined idea and adequate analysis based on the literature in the period (2010-2021).
One suggestion is to continue along this line by explaining in future research whether COVID-19 has been a barrier and/or a driver on the implementation of CSR in emerging and developed countries to know how the pandemic has affected the country under analysis.
Please, check this in page 2:
European Commission (2019). This definition is from 2011.
The European Commission in its Communication (COM (2011) 681) on 25 October 2011 defined Corporate Social Responsibility (CSR) as “the responsibility of enterprises for their impacts on society”.
Author Response
Point 1: It is an adequate work that provides information on drivers and barriers of CSR through a study based on the selection of papers from MENA region. As a consequence, specific drivers and barriers are provided in these countries. So, the knowledge about CSR in emerging countries is completed, together with other existing literature, in comparison with the drivers and barriers in developed countries. As a consequence, a broad vision of CSR is described in this paper. The work is therefore timely, the author uses a well-structured information gathering methodology and provides some tables that serve to have a clear idea of the contribution. In short, it is a well-written work with a well-defined idea and adequate analysis based on the literature in the period (2010-2021).One suggestion is to continue along this line by explaining in future research whether COVID-19 has been a barrier and/or a driver on the implementation of CSR in emerging and developed countries to know how the pandemic has affected the country under analysis.
Response 1: Thank you very much for your constructive feedback, I really appreciate your compliments and recommendations. In order to address your point. I added one paragraph in the conclusion section and elaborated more about future research. I hope my edits have addressed your concern.
Point 2: Please, check this in page 2: European Commission (2019). This definition is from 2011.
The European Commission in its Communication (COM (2011) 681) on 25 October 2011 defined Corporate Social Responsibility (CSR) as “the responsibility of enterprises for their impacts on society”.
Response 2: Thank you very much for your comments I changed 2019 to 2011.

Reviewer 2 Report
The article deals with a very interesting topic: the influence of culture and geographic location on the corporate social responsibility developed by the company.
The authors review 28 articles. Initially, it seemed to me a small number, but finally they justify this number, due to the scarce existing research in these geographical areas.
I find the conclusions interesting, but I would like to know where the authors would direct future research.
As a minor revision I would comment that some citations in the text lack the year or do not follow the format of the journal.
Best regards
Author Response
Point 1: The article deals with a very interesting topic: the influence of culture and geographic location on the corporate social responsibility developed by the company.
The authors review 28 articles. Initially, it seemed to me a small number, but finally they justify this number, due to the scarce existing research in these geographical areas.
I find the conclusions interesting, but I would like to know where the authors would direct future research. As a minor revision I would comment that some citations in the text lack the year or do not follow the format of the journal.
Response 1: Thank you very much for your time and constructive feedback, I really appreciate your guidance. I added few more sentences and talked about future research possibilities in the conclusion section. I also checked the citations and made some changes. I will make sure to ask the journal editor about the citations and Sustainability journal format.

Reviewer 3 Report
I want to thank the author for the work undertaken and for the opportunity to review it. In the piece "The drivers and barriers of corporate social responsibility: A comparison of the MENA region and Western European countries" the author conducts a literature review to try to identify the similarities and differences between those elements that drive CSR and those that hinder it between MENA and Western Europe.
There are many grammatical problems that need to be cleaned up throughout the paper. For instance, in the very first sentence of the abstrate, there should be a "been" between have and published. Using expressions or colloquialisms like "nowadays" is vague. There are many verb tense agreement problems throughout the paper.
The title suggests that you are comparing Europe and MENA regions but then you include the U.S. and Australia.
On line 41: what does "organization's core system" mean?
Line 56: I would not include Australia as a "western country" nor would I call European countries "western," they are European.
Methods: Since you are only reviewing the abstracts, why not look through all 406? This does not seem overly taxing and may result in a larger sample size than 19.
Under the "CSR Drivers" section. You mention "they" a lot but should be including the authors names like you do in the following sections.
You should structure your results in a better way.
I think that there is a serious flaw in your study that is based on a faulty assumption. The assumption that you make is that the researchers that are included in your sample study a region to discover the barriers / drivers of CSR in a region and therefore you can compare the papers to determine the differences. I think that is a bad assumption. Rather, I think that the researchers have a theory about the drivers / barriers and they select the context typically because of access to data. Therefore, the studies are not comparable to determine what similarities / differences there are. Someone would have to take a study done in Europe (MENA) and replicate it in MENA (Europe) to determine if they exist or not in both.
I hope that the comments are helpful and wish you much success in your work.
Author Response
Point 1: There are many grammatical problems that need to be cleaned up throughout the paper. For instance, in the very first sentence of the abstrate, there should be a "been" between have and published. Using expressions or colloquialisms like "nowadays" is vague. There are many verb tense agreement problems throughout the paper.
Response 1: Thank you very much for your guidance, I revised the manuscript by following your recommendations. About the word “Nowadays” in line 40, I showed that to my native English speaker friend and he said “Its not wrong to use Nowadays in that sentence”. If you feel I have to change it, let me know. I will follow your suggestion.
One of my native English speaker friends also read the whole article and helped to improve the text.
Point 2: The title suggests that you are comparing Europe and MENA regions but then you include the U.S. and Australia.
Response 2: Thank you very much for your great comment. I deleted the word “European” from the title.
Point 3: On line 41: what does "organization's core system" mean?
Response 1: Thank you very much for your valuable advice. I change that sentence and used a more familiar term.
Point 4: Line 56: I would not include Australia as a "western country" nor would I call European countries "western," they are European.
Response 2: Thank you very much for your recommendation. I quoted this statement from ABC NIWS : “While geographically close to Asia, Australia is a Western nation, proven by the fact that our political and legal institutions and much of our language and literature are derived from Britain and Europe”.
Retrieved from :https://www.abc.net.au/news/2011-01-26/donellyoz/43240#:~:text=While%20geographically%20close%20to%20Asia,derived%20from%20Britain%20and%20Europe.
Other resources that prove Australia is a western country:
https://worldpopulationreview.com/country-rankings/western-countries
https://en.wikipedia.org/wiki/Western_world
There are numerous academic scholars that included Australia in their research as a western country, for instance:
Gibson, P. G., Henry, R. L., Shah, S., Powell, H., & Wang, H. (2003). Migration to a western country increases asthma symptoms but not eosinophilic airway inflammation. Pediatric pulmonology, 36(3), 209-215.
Tawk, H. M., Vickery, K., Bisset, L., Selby, W., Cossart, Y. E., & Infection in Endoscopy Study Group. (2006). The impact of hepatitis B vaccination in a Western country: recall of vaccination and serological status in Australian adults. Vaccine, 24(8), 1095-1106.
McAdam, J. (2003). Australia and Europe—Worlds Apart. Briefs, 28(4), 192-195.
If you have any other suggestions, I can change the “western countries” to your suggested word. There are also numerous resources that shows researchers see Europe as a part of western world. (All of the above references also see Europe as a main part of western countries. I prefer to follow those researchers that see Europe and Australia regions parts of “Western world”, Unless you all disagree. Thank you very much.
Point 5: Methods: Since you are only reviewing the abstracts, why not look through all 406? This does not seem overly taxing and may result in a larger sample size than 19.
Response 5: Thank you very much for sharing your suggestion. I appreciate it, but I think its more practical for Meta-analysis or bibliometric studies. ONLY for the first step, I reviewed the abstracts not for my literature review. For a structured literature review, I have to read the whole paper. Let's imagine I read all of the 406 and I found 170 relevant articles from western countries. There are only 9 articles for the MENA region, and I don’t find it rational to compare hundreds of articles with a sample of nine. In addition, I used a well-known method to reduce the number of samples (for the time that we have to deal with large sample sizes).
Point 6: Under the "CSR Drivers" section. You mention "they" a lot but should be including the authors names like you do in the following sections. You should structure your results in a better way.
Response 6: Thank you very much for your valuable guidance I checked the Corporate Social Responsibility Drivers section and reduced the number of word “they”. I am not clear what do you mean by “structuring the results in a better way”. With all do respect, I would prefer to follow the other reviewers for the results section.
Point 7: I think that there is a serious flaw in your study that is based on a faulty assumption. The assumption that you make is that the researchers that are included in your sample study a region to discover the barriers/drivers of CSR in a region and therefore you can compare the papers to determine the differences. I think that is a bad assumption. Rather, I think that the researchers have a theory about the drivers/barriers and they select the context typically because of access to data. Therefore, the studies are not comparable to determine what similarities/differences there are. Someone would have to take a study done in Europe (MENA) and replicate it in MENA (Europe) to determine if they exist or not in both.
Response 7: Thank you very much for your great recommendation. I added the limitation section and described the limitation of my study. You mentioned that “researchers have a theory about the drivers / barriers” but I could not find any theory. There are numerous well-established studies that looked for drivers and barriers of different variables (e.g. employee engagement, employee trust, motivation, online shopping, learning) and compared different countries, industries or regions. This is a general comparison because there is very limited data about drivers and barriers of CSR in the MENA region. This study can start the discussion and encourage more research on CSR in that specific region. For example: western countries have more focus on oil and gas companies while none of the MENA studies looked at the Oil and Gas. This is important because this industry is a significant source of emissions on our planet.

Reviewer 4 Report
Dear author,
I had the opportunity to read your article. It follows from the article that you have rather focused on a theoretical overview of the issues addressed.
In my view, such a detailed review is an excellent basis for a scientific article. On the other hand, if I take into account that the article should remain as a review of the literature, I would expect a more detailed discussion, where the author would confront the individual currents of opinion. That is why, in my view, this article is not yet suitable for publication.
Although I now give a negative opinion, I think that this article can be used as an excellent basis for the future when the author enriches it with own research or knowledge.
Author Response
Point 1: I had the opportunity to read your article. It follows from the article that you have rather focused on a theoretical overview of the issues addressed. In my view, such a detailed review is an excellent basis for a scientific article. On the other hand, if I take into account that the article should remain as a review of the literature, I would expect a more detailed discussion, where the author would confront the individual currents of opinion. That is why, in my view, this article is not yet suitable for publication. Although I now give a negative opinion, I think that this article can be used as an excellent basis for the future when the author enriches it with own research or knowledge.
Response 1: Thank you very much for your comments and recommendations. I tried to add more information in the conclusion, recommendation, and limitation sections. I believe the discussion section has enough ideas but the conclusion section in specific needed to have more information. Please let me know what do you think about the new version of my paper. Thanks.

Reviewer 5 Report
Originality - The topic of the article is relevant and interesting, as little is known about CSR in MENA countries.
Literature review - The article summarises the different definitions of CSR but does not describe which one the author accepts. The MENA countries are defined differently by different organisations and even the number of them is different. The article does not describe which definition is adopted and why. MENA is a geographic category; it would be worthwhile to describe the similarities and differences between countries based on some characteristics and justify why the Author thinks he can define common characteristics for this group of countries from a CSR perspective.
Research design – The author's aim was to identify CSR drivers and barriers for Western and MENA countries. It is not clear why the Author thinks that Western countries or MENA countries are uniform in this respect. Are the CSR drivers and barriers really the same in the US and Europe? Or is Europe, for example, uniform in this respect? It would be worthwhile to answer these questions in the article or to point out the limitations.
Method – The author used a structured literature review method, which is a relevant and accepted way to review the literature.
Sample The author examined 28 articles, which is not a large sample in a literature review. It may be worth reconsidering the narrowing down. E.g. "Abstract must show clear indication of drivers or barriers of corporate social responsibility". The author justifies this by saying that "The focus of the research is to study drivers and barriers of corporate social responsibility", but there may be articles on CSR whose abstract does not contain the words driver and barrier, yet the content of the article is related to them. I would suggest for consideration extending the sample, at least for MENA countries, to broader CSR articles.
Results
The narrow sample also leaves the reader with a sense of incompleteness about the results. For example, the article refers to religious belief as one of the main CSR drivers in MENA countries, but does not mention the philanthropic dimension, which may be a pronounced driver in MENA countries (see e.g. Alhares et la, 2021 https://doi.org/10.22495/jgrv10i4art1).
Author Response
Point 1: Originality - The topic of the article is relevant and interesting, as little is known about CSR in MENA countries.
Response 1: Thank you very much for your supportive comment, I really appreciate it.
Point 2: Literature review - The article summarises the different definitions of CSR but does not describe which one the author accepts. The MENA countries are defined differently by different organisations and even the number of them is different. The article does not describe which definition is adopted and why. MENA is a geographic category; it would be worthwhile to describe the similarities and differences between countries based on some characteristics and justify why the Author thinks he can define common characteristics for this group of countries from a CSR perspective.
Response 2: I really appreciate your constructive comments. I edited the manuscript (line 81-83) and clarified my preference among the mentioned definitions. Your comment on the MENA region is also very constructive. I checked different definitions on the MENA region and decided to follow the www.worldatlas.com definition and include 19 countries. (line 105-107).
For the third concern in this section: As I mentioned MENA has some of the world's largest oil and gas reserves and is responsible for the vast amounts of toxic air and water pollution and is one of the largest sources of greenhouse gas emissions in the world”. At the same time, majority of countries does not have any law to mandate CSR activities. I believe there is a need to have more attention on CSR in that region. (In addition, I am originally from that region, studied oil engineering and have some related work experience, this is my interest to study and publish more on this topic).
I think describing similarities and differences between countries in the region is very interesting, but it might change the focus of study. In addition, I have to do the same comparison for western countries. This could an interesting study with different research purpose.
https://www.worldatlas.com/articles/what-are-the-mena-countries.html
Point 3: Research design – The author's aim was to identify CSR drivers and barriers for Western and MENA countries. It is not clear why the Author thinks that Western countries or MENA countries are uniform in this respect. Are the CSR drivers and barriers really the same in the US and Europe? Or is Europe, for example, uniform in this respect? It would be worthwhile to answer these questions in the article or to point out the limitations.
Response 3: Thank you very much for your comment and recommendation. In general, CSR has been studied more in the western countries while we do not see much research related to CSR drivers and barriers in the MENA region. In general, western countries are more developed and have similar education systems, they politically and economically are closer. There is a movement that tries to change organizations to become environmentally friendly. Western countries (in general) are trying to lead the world and reduce the fossil fuel car productions and promote electric cars. I never mentioned these regions are uniform. Even in one specific country, like United states, each industry in each State is different. Organizations can have different drivers or barriers if we aim to focus on details. I just wanted to know what is out there.
I added a limitation section to address this.
Point 4: Method – The author used a structured literature review method, which is a relevant and accepted way to review the literature.
Response 4: Thank you very much for your supportive comment, I really appreciate it.
Point 5: Sample The author examined 28 articles, which is not a large sample in a literature review. It may be worth reconsidering the narrowing down. E.g. "Abstract must show clear indication of drivers or barriers of corporate social responsibility". The author justifies this by saying that "The focus of the research is to study drivers and barriers of corporate social responsibility", but there may be articles on CSR whose abstract does not contain the words driver and barrier, yet the content of the article is related to them. I would suggest for consideration extending the sample, at least for MENA countries, to broader CSR articles.
Response 5: Thank you very much for your suggestions, I agree with you and if I delete "Abstract must show clear indication of drivers or barriers of corporate social responsibility" I will have a lot more articles. But this statement can help me to find articles that clearly aimed to find drivers, antecedents, or barriers of CSR. Only authors that are aiming to find drivers or barriers will have those words in their abstract sections. Actually I didn’t have that limitation that at the beginning but I had to deal with two issues after deleting that statement: 1- The number of articles will increase in both regions, 2- the number of unrelated articles will increase significantly in both regions. It will take a lot more time to analyse hundreds of papers to see how many of them are relevant to the scope of study. I think this can be a great bibliometric study to analyse the papers with VOSViewer software.
I also added this concern in the limitation section.
Point 6: Results: The narrow sample also leaves the reader with a sense of incompleteness about the results. For example, the article refers to religious belief as one of the main CSR drivers in MENA countries, but does not mention the philanthropic dimension, which may be a pronounced driver in MENA countries (see e.g. Alhares et la, 2021 https://doi.org/10.22495/jgrv10i4art1).
Response 6: Thank you very much for sharing this interesting article. I was trying to find why organizations participate in CSR programs; so in the articles that I have collected, the drivers and barriers of CSR have been identified by analyzing organizations’ documents or by surveying the companies’ business leaders, managers, employees, and experts in the field of CSR. Therefore, none of their findings represent consumers perspective. In the paper that you shared Alhares et la surveyed consumers (so its consumers’ perspective). Its not necessarily the underlying reason of CSR implementation.

Round 2
Reviewer 4 Report
Dear author, I appreciate your prompt efforts to correct the article. Unfortunately, from my point of view, there has been no reworking as appropriate. It is still an article that is essentially a review of research conducted by other authors. I do not see such a contribution to science there that would significantly enrich the knowledge of readers from my point of view.
I'm sorry I can't change my mind. Anyway, I'm keeping my fingers crossed for the future.
Reviewer 5 Report
The Author has made several additions to the article and Limitations has included the parts that were objected to in the Reviews.